# A human forebrain organoid model reveals the essential function of GTF2IRD1-TTR-ERK axis for the neurodevelopmental deficits of Williams syndrome

Xingsen Zhao[1,2,3†], Qihang Sun[1,2†], Yikai Shou[1†], Weijun Chen[1†], Mengxuan Wang[1,2], Wenzheng Qu[1], Xiaoli Huang[1], Ying Li[1], Chao Wang[4], Yan Gu[4], Chai Ji[1*], Qiang Shu[1*], Xuekun Li[1,2,3*]

[1]The Children's Hospital, National Clinical Research Center for Child Health, School of Medicine, Zhejiang University, Hangzhou, China; [2]The Institute of Translational Medicine, School of Medicine, Zhejiang University, Hangzhou, China; [3]Binjiang Institute of Zhejiang University, Hangzhou, China; [4]Center of Stem Cell and Regenerative Medicine, and Department of Neurology of the Second Affiliated Hospital, Zhejiang University School of Medicine, Hangzhou, China

*For correspondence:
6198011@zju.edu.cn (CJ);
shuqiang@zju.edu.cn (QS);
xuekun_li@zju.edu.cn (XL)

†These authors contributed equally to this work

## eLife assessment

Zhao et al. report **valuable** adverse effects on cell proliferation, differentiation and gene expression, possibly linked to reduced binding activity of the transcription factor GTF2IRD1 to the transthyretin (TTR) promoter, in a human forebrain organoid model of Williams Syndrome (WS). The authors provide **incomplete** evidence of the effects of GTF2IRD1, a mutated gene in WS, on altering MAPK/ERK pathway activity, a well-recognized target in cell proliferation.

**Abstract** Williams syndrome (WS; OMIM#194050) is a rare disorder, which is caused by the microdeletion of one copy of 25–27 genes, and WS patients display diverse neuronal deficits. Although remarkable progresses have been achieved, the mechanisms for these distinct deficits are still largely unknown. Here, we have shown that neural progenitor cells (NPCs) in WS forebrain organoids display abnormal proliferation and differentiation capabilities, and synapse formation. Genes with altered expression are related to neuronal development and neurogenesis. Single cell RNA-seq (scRNA-seq) data analysis revealed 13 clusters in healthy control and WS organoids. WS organoids show an aberrant generation of excitatory neurons. Mechanistically, the expression of transthyretin (TTR) are remarkably decreased in WS forebrain organoids. We have found that GTF2IRD1 encoded by one WS associated gene *GTF2IRD1* binds to *TTR* promoter regions and regulates the expression of *TTR*. In addition, exogenous TTR can activate ERK signaling and rescue neurogenic deficits of WS forebrain organoids. *Gtf2ird1*-deficient mice display similar neurodevelopmental deficits as observed in WS organoids. Collectively, our study reveals critical function of GTF2IRD1 in regulating neurodevelopment of WS forebrain organoids and mice through regulating TTR-ERK pathway.

## Introduction

Williams syndrome, also known as Williams-Beuren syndrome (WS; OMIM#194050) is a rare disorder with a prevalence of 1 in 7500 live births (*Kozel et al., 2021*; *Pober, 2010*). WS is caused by the microdeletion of approximately 25–27 genes (termed as WS genes) on chromosome 7q11.23 and affects multiple systems including cardiovascular and neuronal systems (*Collins, 2013*; *Collins, 2018*; *Lin et al., 2019*; *Pober et al., 2008*). Most individuals with WS displayed neuronal phenotypes including hypersociability, impaired myelination and intellectual disability (IQ <70; *Barak and Feng, 2016*; *Barak et al., 2019*; *Kozel et al., 2021*; *Morris and Braddock, 2020*). fMRI neuroimaging and examination with post-mortem brain tissues of WS patients both showed abnormalities of brain structure of WS patients, including visuospatial construction deficit and increased length of dendrites, numbers of spine and branching points (*Chailangkarn et al., 2016*; *Kippenhan et al., 2005*; *Meyer-Lindenberg et al., 2004*).

Neural progenitor cells (NPCs) and neurons derived from the induced pluripotent stem cells (iPSC) of WS patient display deficits, such as increased apoptosis, aberrant morphological complexity and electrophysiological activity (*Chailangkarn et al., 2016*; *Khattak et al., 2015*). The dysregulation of Wnt signaling pathway plays important roles in regulating the survival and development of neurons derived from induced pluripotent stem cells (iPSCs) of WS patient. Neuronal deletion of WS gene Gtf2i leads to severe behavioral deficits and reduced myelination in mice (*Barak et al., 2019*). Although single gene mutation of those 25–27 genes only causes partial phenotypes of WS, genotype-phenotype relationship analysis suggests that *GTF2IRD1*, *GTF2I*, BAZ1B, VPS37D, *STX1A*, *CLIP2*, *LIMK1*, *MLXIPL,* and *ELN* could be the key genes for the phenotypes (*Kozel et al., 2021*; *Lek et al., 2016*). GTF2IRD1 encodes DNA binding protein of the Transcription factor (TF) II-I family and *its* deficiency contributes to neurological deficits of WS patients, such as the impaired motor coordination, visual-spatial constructive deficits and cognitive deficits (*Carmona-Mora et al., 2015*; *Corley et al., 2016*; *Dai et al., 2009*; *Enkhmandakh et al., 2009*; *Kopp et al., 2020*; *Makeyev and Bayarsaihan, 2009*; *Morris et al., 2003*; *O'Leary and Osborne, 2011*; *Tassabehji et al., 2005*). Despite this remarkable progress, the mechanisms for diverse neuronal defects of WS still remain largely unknown.

Due to the limited accessibility of human brain samples of WS patients, diverse cellular and animal models have been used to study the phenotypes and mechanisms of WS. Brain organoids are derived iPSCs under 3D cultural condition and exhibit unique advantages in modeling spatio-temporal features of brain development compared to 2D cultures including iPSCs. Therefore, brain organoids have been applied for studying human brain development and neurological disorders, including Fragile X syndrome and autism (*Clevers, 2016*; *Di Lullo and Kriegstein, 2017*; *Kang et al., 2021a*; *Kelava and Lancaster, 2016*; *Li and Izpisua Belmonte, 2019*; *Lyon, 2019*; *Paşca, 2019*; *Qian et al., 2019*). To our knowledge, there have been no studies performed with brain organoids of WS.

In present study, utilizing the iPSCs derived forebrain organoids, we found that NPCs in WS forebrain organoids displayed abnormal proliferation and differentiation, and synapse formation. Bulk RNA-seq revealed the altered gene expression in WS organoids, which is strongly related to neuronal development and neurogenesis. scRNA-seq data analysis revealed 13 clusters in healthy control and WS organoids, and WS NPCs showed an aberrant excitatory neurogenesis. Mechanistically, the expression of transthyretin (TTR) significantly decreased in WS forebrain organoids. GTF2IRD1 encoded by one WS gene *GTF2IRD1* binds to *TTR* promoter regions and regulates the expression of *TTR*. Exogenous TTR can rescue neurogenic and neurodevelopmental deficits of WS forebrain organoids by activating ERK signaling. Taken together, our results reveal the critical neurogenesis features of WS forebrain organoids and provide novel insight into mechanisms underlying the abnormal neurodevelopment of WS.

## Results

### The forebrain organoids of William's syndrome patient display deficits of neurogenesis and neuronal differentiation

To study the neuronal deficits in WS patient, we isolated peripheral blood mononuclear cells (PBMCs) and urinary cells from healthy control (CTRL) and WS patients (*Figure 1—figure supplement 1A*), respectively. After the clinical evaluations of phenotypes of three WS patients (*Figure 1—figure supplement 1B*), target sequencing was performed and showed the hemizygous deletion of 23 genes

in three WS patients (*Figure 1—figure supplement 1C and D*). PBMCs and urinary cells of CTRL and WS patients were cultured in vitro, respectively (*Figure 1—figure supplement 2A*). For reprogramming, cultured cells were infected with Sendai virus according to the manufacturer's protocol. Three induced pluripotent stem cells (iPSCs) lines were characterized for each CTRL and WS samples. iPSCs displayed typical morphology, normal karyotypes, were positive for alkaline phosphatase live staining and expressed canonical pluripotency markers SOX2, NANOG, SSEA4, and OCT4 (*Figure 1—figure supplement 2B–F*).

To assess the neurodevelopmental deficits of WS patients, we generated three dimensional (3D) forebrain organoids with CTRL and WS iPSCs using an established protocol, respectively, which could be well correlated with fetal human brain development (*Figure 1A*; *Kang et al., 2021a*; *Qian et al., 2016*). Considering day 56 (D56) forebrain organoids were well correlated with fetal brain development (*Kang et al., 2021a*; *Kostović et al., 2019*; *Qian et al., 2016*), our following experiments adopted D56 forebrain organoids if no specific annotation. We first performed immunofluorescence staining and observed that WS forebrain organoids showed a higher percentage of KI67$^+$/SOX2$^+$ cells compared to that of CTRL organoids (*Figure 1B and C*). Further, CTRL and WS organoids were pulsed with 5-ethynyl-2'-deoxyuridine (EdU) at a dosage of 10 µM for 2 hr, and we found that the percentage of EdU$^+$/SOX2$^+$ cells also significantly increased in WS organoids (*Figure 1D and E*). Furthermore, CTRL and WS organoids were incorporated with EdU for 24 hr, and the cell cycle exiting of neural progenitor cells (NPCs) was assessed by analyzing the co-localization of EdU and Ki67. We observed that the percentage of Ki67$^-$EdU$^+$ cells was significantly decreased in WS organoids (*Figure 1F and G*). Fewer TBR2$^+$ intermediate neural progenitor cells (IPCs) localized at MAP2$^-$ ventricular zone (VZ) like layer in WS forebrain organoids compared to the CTRL organoids (*Figure 1H and I*). These results suggest an aberrant proliferating capability of NPCs in WS forebrain organoids.

Next, we examined the differentiation capability of NPCs. We found that upon the differentiation, the expression of neuronal precursor and immature neuronal cell marker doublecortin positive (DCX$^+$) cells at SOX2$^+$ ventricle (VZ)-like layer significantly decreased in WS forebrain organoids (*Figure 2A and B*). The relative thickness of the VZ layer indicated by SOX2$^+$ aberrantly increased, but the thickness of cortical plate (CP) layer indicated by MAP2$^+$ abnormally decreased (*Figure 2*). The total number (*Figure 2F and G*) and the proporation of neuronal marker CTIP2$^+$ cells in CP like layer (*Figure 2F and H*) were significantly decreased in WS forebrain organoids. The total number (*Figure 2I and J*) and the proporation of neuronal marker TBR1$^+$ cells in CP like layer (*Figure 2I and K*) were also significantly decreased in WS forebrain organoids. These results suggest a dysregulated neuronal differentiation of NPCs in WS forebrain organoids.

## WS forebrain organoids display the aberrant expression of neurodevelopmental genes

Next, we performed bulk RNA sequencing (RNA-seq) with both CTRL and WS forebrain organoids. In CTRL forebrain organoids. *Figure 3A* showed the FPKM values of 23 WS-associated genes, and other three genes *LAT2*, *TREM270* and *FKBP6* showed very low expression (FPKM less than 1). The expression of these 23 WS genes significantly decreased in WS forebrain organoids (*Figure 3B*). In addition, RNA-seq data analysis revealed 1761 differentially expressed genes (DEGs): 1003 down-regulated and 758 up-regulated (*Figure 3C*, *Supplementary file 1*). Gene Ontology (GO) analysis with all DEG showed enrichment for neuronal development and neurogenesis (*Figure 3D*), and interaction network of top 30 genes related to neurogenesis was shown (*Figure 3E*). GO analysis also revealed the enrichment of up-regulated genes for cell proliferation and of down-regulated genes for neuronal differentiation, synaptic signaling and neurogenesis (*Figure 3F and G*), respectively. In addition, FPKM and qRT-PCR results showed the decreased levels of NPC markers *SOX2*, *NESTIN*, *PAX6*, and *HES5* (*Figure 3H*, *Figure 3—figure supplement 1A*) and the increased levels of neuronal differentiation markers, such as *TBR1*, *TBR2*, *CTIP2*, and *SATB2* (*Figure 3I*, *Figure 3—figure supplement 1B*) in WS organoids. WB assay and quantification results showed that the levels of SOX2, NESTIN, PAX6, and HES5 were significantly increased (*Figure 3J–N*) and the levels of TBR1, SATB2, and CTIP2 (*Figure 3O–R*) were significantly decreased in WS organoids compared to CTRL. Collectively, these data suggested the aberrant expression of neurogenic and neurodevelopmental genes in WS forebrain organoids.

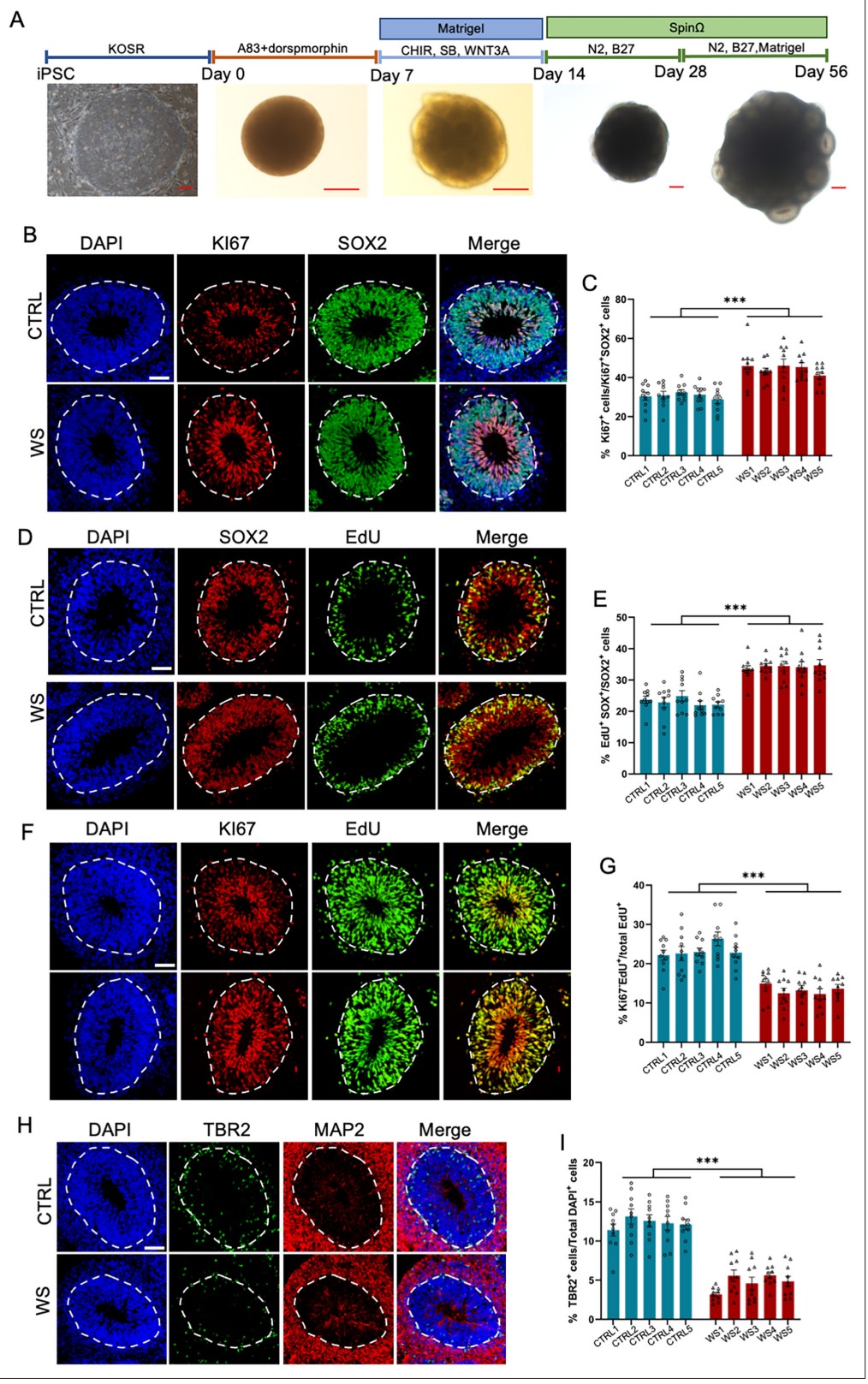

**Figure 1.** WS neural progenitor cells display abnormal proliferating capability. (**A**) Schematic illustration of forebrain organoids culture process and representative brightfield images of organoids over development. Scale bar, 200 μm. (**B**) Representative images of KI67 and neural progenitor cells (NPCs) marker SOX2 immunofluorescence staining with healthy (CTRL) and WS patient (WS) forebrain organoids at day 56, respectively.

*Figure 1 continued on next page*

*Figure 1 continued*

The proliferating cells were positive for KI67 labeling, and NPCs were positive for SOX2 labeling. The white dotted line indicated the outer interface of the ventricular zone (VZ) region. (**C**) Quantification results showed that the average proportion of KI67 positive (KI67$^+$) cells among total NPCs (SOX2$^+$) significantly increased in the VZ layer of WS forebrain organoids relative to CTRL. (**D**) Representative images of EdU and SOX2 immunofluorescence staining with CTRL and WS forebrain organoids at day 56. EdU was applied for 2 hr. (**E**) Quantification results showed that the average proportion of EdU$^+$ cells among total SOX2$^+$ NPCs significantly increased in the VZ layer of WS forebrain organoids compared to CTRL. (**F**) Representative images of EdU and KI67 immunofluorescence staining with CTRL and WS forebrain organoids at day 56, respectively. EdU was applied for 24 hr. (**G**) Quantification results show that the average proportion of Ki67$^-$EdU$^+$ cells among total EdU$^+$ cells significantly decreased in the VZ layer of WS forebrain organoids compared to CTRL. EdU was applied for 24 hr. Ki67$^-$EdU$^+$ cells were counted, that is, the proportion of cells exiting the cell cycle. (**H**) Representative images of TBR2, a marker for intermediate precursor cells (IPCs), and neuronal marker MAP2 immunofluorescence staining of CTRL and WS forebrain organoids, respectively. (**I**) Quantification results showed that the average proportion of TBR2$^+$ IPCs remarkably decreased in WS forebrain organoids compared to CTRL. For (**B-I**), five IPS lines of three healthy control (Ctrl) and three WS patients (WS) were adopted for the study, respectively. Three to five organoids were analyzed from each IPS line and 10 representative rosettes of organoids were analyzed in each line. values represent mean ± SEM. *p<0.05, **p<0.01, ***p<0.001, one-way ANOVA followed by Dunnett's multiple-comparison test. Scale bar, 50 μm.

The online version of this article includes the following figure supplement(s) for figure 1:

**Figure supplement 1.** Clinical information and genetic analysis of CTRL and WS donors.

**Figure supplement 2.** Characterization of primary cells and the generated iPSCs.

## scRNA-seq reveals the abnormal generation of excitatory neurons in WS forebrain organoids

To gain further insight into the specific transcriptome alterations in different cell types of WS, we performed 10 X genomics chromium single-cell RNA-Seq (scRNA-Seq) with WS and CTRL forebrain organoids. After quality control, total 96,969 cells were obtained from CTRL and WS samples, and were analyzed together using unsupervised clustering (*Figure 4A*, *Figure 4—figure supplement 1A*). Uniform manifold approximation and projection (UMAP) identified 13 cell clusters in CTRL and WS forebrain organoids, which were then annotated using the expression of cell-type-specific markers (*Figure 4B*, *Figure 4—figure supplement 1B, C*). We observed that the proportions of deeper layer_ projection neuron (DL_PN) and upper layer projection neuron (UL_PN) clusters were significantly decreased in WS forebrain organoids compared to CTRL, while the proportions of other clusters did not show significant difference between CTRL and WS organoids (*Figure 4C–E*, *Figure 4—figure supplement 1D*). According to the markers for the cluster characterization including *Bcl11B* and *FEZF2*, DL_PN and UL_PN were excitatory neurons (*Paulsen et al., 2022*). scRNA-seq data also revealed the differentially expressed genes (DEGs) in 6 distinct clusters including cycling progenitor, oRG, IPC, newborn DL_PN, DL_PN, and UL_PN, which were related to the development of excitatory neurons (*Figure 4F*). GO analysis with DEGs in DL_PN and UL_PN clusters showed the enrichment for neuronal development and neuronal differentiation (*Figure 4G and H*). Further, GO analysis with DEGs in cycling progenitor, oRG, IPC and new born_DL_PN clusters showed the enrichment for neurogenesis, cell division and cell migration terms, etc (*Figure 4—figure supplement 1E–I*). In particular, UMAP distribution maps and heatmaps of WS-related genes showed a distinct expression pattern of WS-deleted genes in cell clusters (*Figure 4I*). Collectively, these results suggested the deficits of excitatory neurogenesis in WS forebrain organoids.

Given the aberrant production of the excitatory neurons in WS forebrain organoids, we next analyzed the developmental pesudotime trajectories for cycling progenitor, oRG, IPC, newborn DL_PN, DL_PN, and UL_PN clusters of CTRL and WS organoids (*Figure 5A and B*). The pesudotime of excitatory neuronal trajectories in WS organoids analysis showed a remarkable delayed development compared to CTRL organoids (*Figure 5C*). Pseudotime uniform manifold approximation analysis revealed that the excitatory neuronal lineage displayed an increased distribution towards the start point trajectory and a decreased distribution towards the end point trajectory in WS organoids (*Figure 5D*). The varieties of gene expression along pesudotime trajectories was shown by heatmap (*Figure 5E*). Furthermore, the expression of NPC marker genes *SOX2* and *TBR2* exhibited increasing trend at early stage

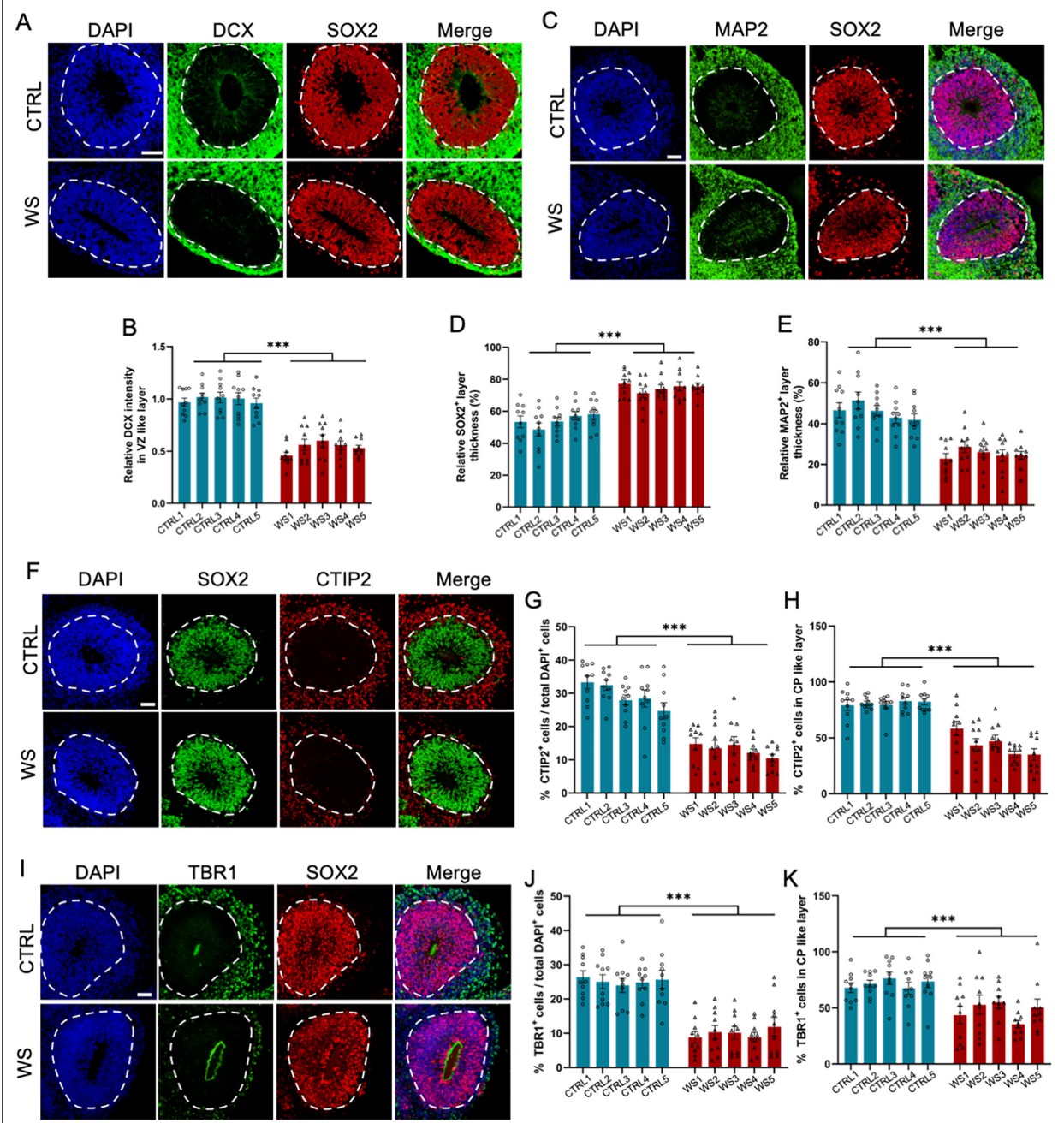

**Figure 2.** WS forebrain organoids exhibit the reduced neuronal differentiation. (**A**) Representative images of DCX and SOX2 immunofluorescence staining of CTRL and WS forebrain organoids, respectively. (**B**) Quantification results showed that the intensity of DCX signal significantly decreased in SOX2+ VZ like layers of WS forebrain organoids compared to CTRL. (**C**) Representative images of MAP2 and SOX2 immunofluorescence staining of CTRL and WS forebrain organoids, respectively. (**D, E**) Quantification results showed that the thickness of VZ like layer (SOX2+MAP2−) significantly increased (**D**), but the thickness of cortical plate (CP) like layer (MAP2+) CP like layer significantly decreased (**E**) in WS forebrain organoids compared to CTRL. (**F**) Representative images of SOX2 and neuronal marker CTIP2 immunofluorescence staining of CTRL and WS forebrain organoids, respectively. (**G**) Quantification results showed that the proportion of CTIP2+ neurons significantly decreased in WS forebrain organoids compared to CTRL. (**H**) Quantification results showed that the relative thickness of SOX2+ VZ like layer significantly increased, while the the relative thickness of CTIP2+ CP like layer significantly decreased in WS forebrain organoids compared to CTRL. (**I**) Representative images of SOX2 and neuronal marker TBR1 immunofluorescence staining of CTRL and WS forebrain organoids, respectively. (**J**) Quantification results showed that the proportion of total TBR1+ neurons significantly decreased in WS forebrain organoids compared to CTRL. (**K**) Quantification results showed that the proportion of TBR1+ neurons significantly decreased in TBR1+ CP like layer of WS forebrain organoids compared to CTRL. For (**A–K**), five IPS lines of Ctrl and WS were adopted for the study, respectively. Three to five organoids were adopted from each IPS line and 10 representative rosettes of organoids were analyzed in each line. values represent mean ± SEM. *p<0.05, **p<0.01, ***p<0.001, one-way ANOVA followed by Dunnett's multiple-comparison test. Scale bar, 50 µm.

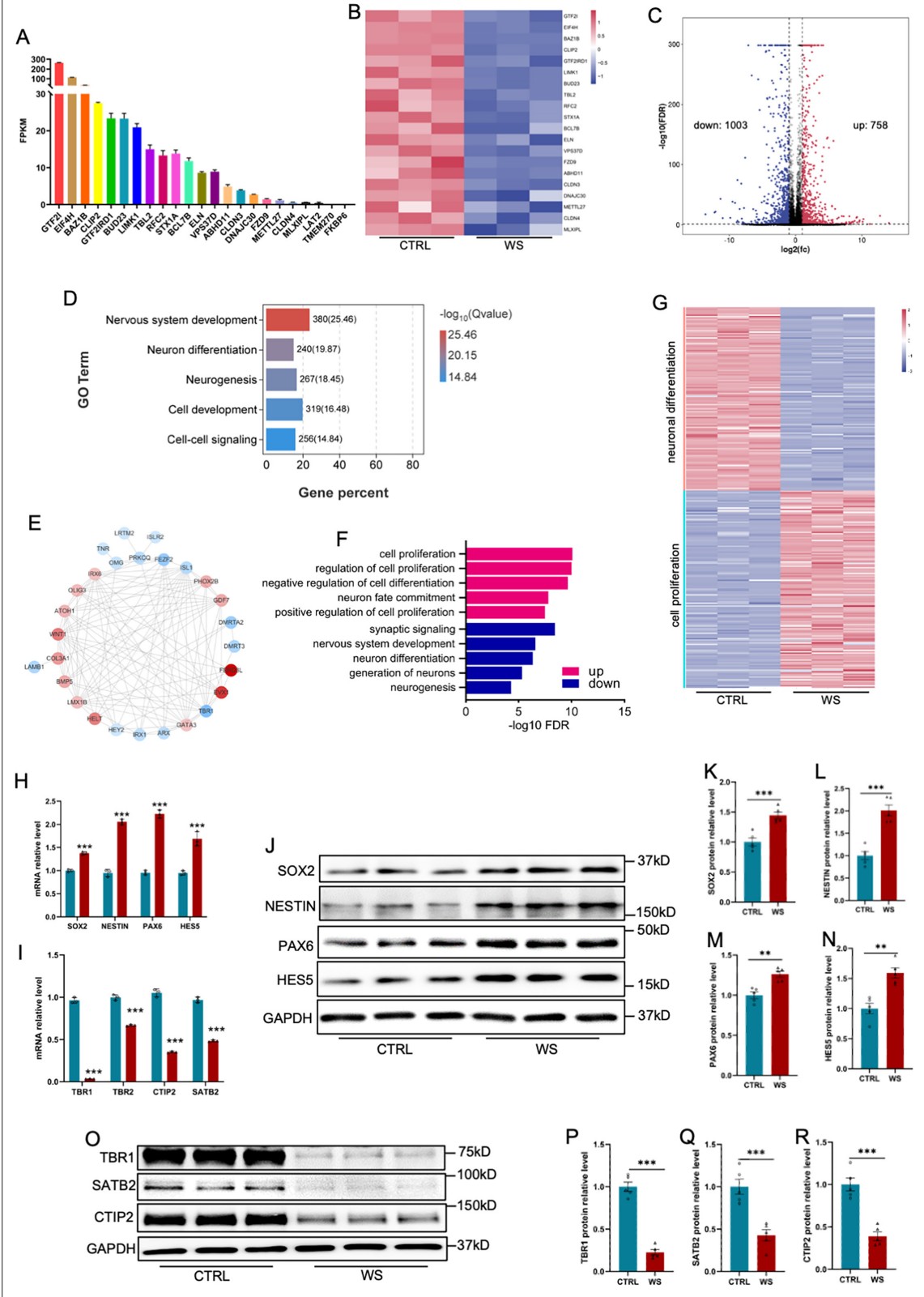

**Figure 3.** WS forebrain organoids display the altered expression of neurogenic genes. (**A**) Fragments per kilobase of transcript per million mapped reads (FPKM) values of 23 WS genes in Day 56 CTRL organoids. (**B**) Heatmap analysis of 23 WS genes in both CTRL and WS brain organoids. (**C**) Volcano plot of the differentially expressed genes (DEGs). Bulk RNA sequencing (bulk RNA-seq) revealed 758 up-regulated and 1003 down-regualted genes in WS brain organoids compared with CTRL organoids, respectively. Fold change >2 and p-value <0.05 were considered as significance. (**D**) Gene

*Figure 3 continued on next page*

*Figure 3 continued*

Ontology (GO) analysis of DEGs showed the enrichment in terms related to neuronal differentiation and neurogenesis. (**E**) Interactome plot of top 27 genes of neurogenesis GO term was presented. Red dots representing up regulated genes and blue dots representing down regulated genes. (**F**) GO analysis of DEGs in WS organoids. Red bar represents terms enriched by up regulated genes, and blue bar represents terms enriched by down regulated genes. (**G**) Heatmap of cell proliferation and neuronal differentiation related genes in CTRL and WS organoids. (**H, I**) qRT-PCR results showed the increased expression of NPCs markers including SOX2, NESTIN, PAX6, and HES5 (**H**) and the decreased expression of neuronal markers including TBR1, TBR1, CTIP2, and SATB2 (**I**) in WS organoids compared to Ctrl group. Values represent mean ± SEM; *p<0.05, **p<0.01, ***p<0.001; unpaired Student's t test. (**J–N**) WB assay (**J**) and quantification results shoed the increased expression of SOX2 (**K**), NESTIN (**L**), PAX6 (**M**), HES5 (**N**) in WS organoids compared to CTRL group. Values represent mean ± SEM; *p<0.05, **p<0.01, ***p<0.001; unpaired Student's t test. (**O–R**) WB assay (**O**) and quantification results of TBR1 (**P**), CTIP2 (**Q**), SATB2 (**R**) in WS organoids compared to CTRL group. Values represent mean ± SEM; n=5 independent experiments; *p<0.05, **p<0.01, ***p<0.001; unpaired Student's t test.

The online version of this article includes the following source data and figure supplement(s) for figure 3:

**Source data 1.** PDF file for original blots images in *Figure 3J and O*.

**Source data 2.** TIF files for original blots images in *Figure 3J and O*.

**Figure supplement 1.** Bulk RNA sequencing (RNA-seq) reveals the altered expression of neurodevelopmental genes.

of trajectories, but the expression of neuronal marker genes *NHLH1*, *TBR1*, *CTIP2*, and *SATB2* showed decreasing trend at middle and later stages of trajectories in WS organoids (*Figure 5F*). Collectively, these data suggested an aberrant neurodevelopmental timeline in WS organoids.

## The deficiency of *GTF2IRD1* reduces *Transthyretin* in WS brain organoids

Next, we aim to reveal the mechanism underlying the neuronal deficits of WS organoids. Integrated analysis with down-regulated genes identified both by bulk RNA-seq data and scRNA-seq, respectively, revealed one overlapped gene *transthyretin* (*TTR*; *Figure 6A*), whereas there is no overlapped genes were identified in up-regulated genes by bulk RNA-seq data and scRNA-seq, respectively (*Figure 6—figure supplement 1A*). TTR is required for the transportation of thyroxine and retinol and plays an essential function in the development of the central nervous system (*Fleming et al., 2009*; *Gomes et al., 2016*; *Magalhães et al., 2021*; *Santos et al., 2010*). Interestingly, a previous study has showed a decreased expression of *TTR* in the brain of WS patients (*Figure 6—figure supplement 1B*; *Barak et al., 2019*). scRNA-seq data analysis showed that *TTR* was widely expressed in all cell clusters of CTRL organoids, but significantly decreased in WS organoids (*Figure 6B and C*). Immunofluorescence staining, qRT-PCR and WB assay results confirmed the remarkable decrease of TTR in WS brain organoids relative to Ctrl (*Figure 6D–G*, *Figure 6—figure supplement 1C*). Of note, RNA-seq data anaylsis showed that down-regulated genes enriched for MAPK signaling pathway (*Figure 6H*). WB assay results showed the significant reduction of GTF2IRD1 and p-ERK/ERK in WS brain organoids (*Figure 6I–L*). These results indicate a reduced expression of TTR and defects of MAPK/ERK signaling in WS brain organoids.

Next, we aim to uncover the mechanism of how TTR was regulated. Two genes among the deleted 23 genes in WS patients, *GTF2I* and *GTF2IRD1*, are two transcription factors and were widely expressed in distinct cell clusters in CTRL organoids revealed by scRNA-seq (as shown in *Figure 4I*). We speculated whether *GTF2I* and *GTF2IRD1* regulated the expression of TTR, and then performed knockdown (KD) of *GTF2I* and *GTF2IRD1* with shRNAs in HEK293T cells, respectively. We observed that the depletion of *GTF2IRD1* but not *GTF2I* led to a significant decrease of *TTR* (*Figure 6—figure supplement 1D and E*). These results collectively suggest that *GTF2IRD1* deficiency indeed reduces the expression TTR.

## Exogeneous TTR rescues the neuronal deficits in WS brain organoids

Next, we aim to examine whether TTR application could rescue the neuronal deficits in WS brain organoids. WS brain organoids at day 28 were exposed to TTR recombinant protein (55 µg/ml) and collected for assays at day 56 (*Figure 7—figure supplement 1A*). WB assay and quantification results showed TTR exposure significantly increased the level p-ERK (*Figure 7A–C*), which could be significantly inhibited by the supplement of TTR inhibitor receptor-associated protein (RAP), a ligand inhibiting the internalization of TTR (*Figure 7D–F*). Both TTR only and TTR plus RAP did not affefct the level of total ERK (*Figure 7C and F*). RAP Immunofluorescence staining and quantification results showed

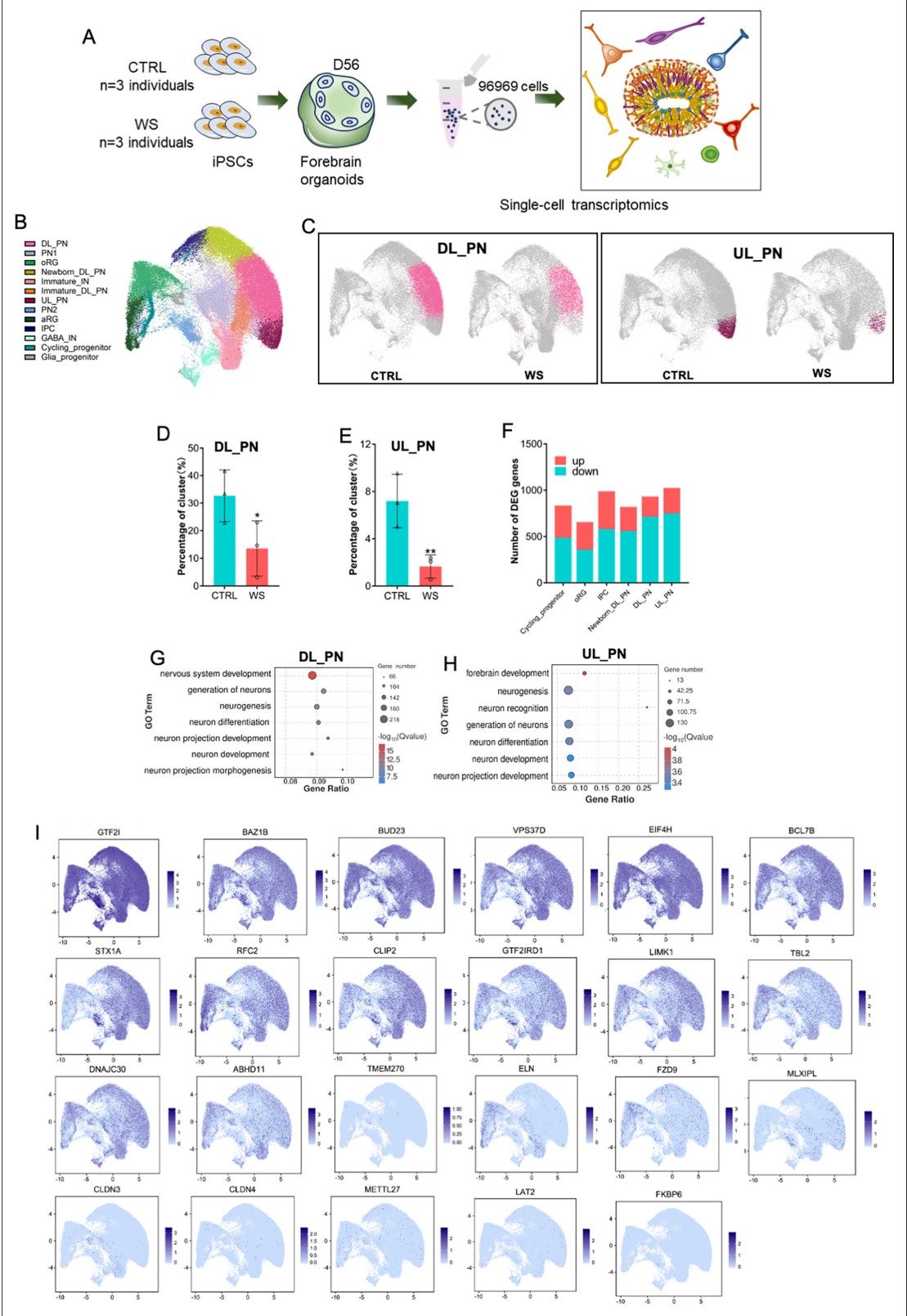

**Figure 4.** scRNA-seq reveals the abnormal neuronal differentiation in WS forebrain organoids. (**A**) Schematic illustration of the scRNA-seq and histological analysis of both CTRL and WS forebrain organoids. (**B**) UMAP plot of cell types detected in both CTRL and WS forebrain organoids. DL_PN, deep layer projection neurons; UL_PN, upper layer projection neurons; IN, interneurons; aRG, apical radial glia; oRG, outer radial glia; IPC, intermediate progenitor cells; GABA IN, GABAergic interneurons. (**C**) UMAP plot of DL_PN (left) and UL_PN (right) in both CTRL and WS forebrain organoids. (**D,**

*Figure 4 continued on next page*

Figure 4 continued

**E**) Quantificationof the percentages of DL_PN and UL_PN in both CTRLcontr and WS forebrain organoids. Values represent mean ± SEM; *p<0.05, **p<0.01, ***p<0.001; unpaired Student's t test. (**F**) Number of DEGs in the indicated cell types in WS organoids. Fold change >1.28 and p-value <0.05 were considered as significance. (**G, H**) GO analysis of DEGs in the DL_PN (**I**) and UL_PN (**J**) cell types in WS organoids. (**I**) UMAP visualization of the expression of WS genes in CTRL forebrain organoids.

The online version of this article includes the following figure supplement(s) for figure 4:

**Figure supplement 1.** scRNA-seq reveals the abnormal neuronal differentiation in WS forebrain organoids.

that both TTR exposure remarkably enhanced the percentages of CTIP2$^+$ cells compared to untreated WS organoids (**Figure 7G and H**). TTR exposure also significantly increased the percentages of TBR1$^+$ cells (**Figure 7I and J**). These results suggest that TTR can rescue the neuronal deficits of WS forebrain organoids.

### *Gtf2ird1* deficiency induces neurodevelopmental deficits in mice

Finally, we aim to examine whether *Gtf2ird1* deficiency also induces neurodevelopmental deficits in mice. *Gtf2ird1$^{+/+}$* (Wild-type, WT), *Gtf2ird1$^{+/-}$* (heterozygous, Het) and *Gtf2ird1$^{-/-}$* (knockout, KO) mice were adopted for the study. Immunofluorescence staining with brain sections and quantification results showed that *Gtf2ird1$^{+/-}$* and *Gtf2ird1$^{-/-}$* mice had the increased BrdU$^+$ (**Figure 8A and B**) and Ki67$^+$Sox2$^+$ (**Figure 8C and D**) neuronal progenitor cells in VZ/SVZ regions, respectively, but had the decreased CTIP2$^+$ (**Figure 8E and F**), STAB$^+$ cells (**Figure 8E and G**) and TBR1$^+$ (**Figure 8H and I**) in IZ/CP regions relative to WT mice, respectively. In addition, WB assay and quantification results showed that the protein levels of Gtf2ird1, p-ERK and Ttr (**Figure 8J–N**) were significantly decreased in the cortex of *Gtf2ird1$^{+/-}$* and *Gtf2ird1$^{-/-}$* mice relative to WT mice, respectively.

Next, we examined the effects of Gtf2ird1 on the proliferation and differentiation of embryonic neural progenitor cells of mouse (eNPCs) in vitro. eNPCs were isolated from the forebrains of WT, Het and KO mice, respectively. BrdU incorpation assay and quantification results showed that the percentage of BrdU$^+$Nestin$^+$/Nestin$^+$ (**Figure 9A and B**) and the percentage of Ki67$^+$Sox2$^+$/Sox2$^+$ (**Figure 9C and D**) were significantly increased in Het and KO eNPCs, respectively. Upon the differentiation, Het and KO eNPCs produced fewer Tuj1$^+$ neurons and more Gfap$^+$ astrocytes relative to WT eNPCs (**Figure 9E–G**), respectively. qRT-PCR results consistently showed the altered expression of *Gtf2ird1, Ttr, Ki67, Nestin,* and *Sox2* in proliferating eNPCs (**Figure 9—figure supplement 1A**) and *Gtf2ird1, Ttr, Tuj1,* and *Gfap* (**Figure 9—figure supplement 1B**) in differentiated eNPCs, respectively. We then further performed acute knockdown of *Gtf2ird1* (KD) in eNPCs and found that *Gtf2ird1* KD significantly reduced the mRNA level of *Ttr* (**Figure 9J–L**). WB assay and quantification results showed thaht *Gtf2ird1* KD also significantly reduced the protein level of Ttr (**Figure 9J–L**). In addition, *Gtf2ird1* KD significantly inhibited neuronal differentiation of mouse eNPCs (**Figure 9—figure supplement 1C and D**).

Finally, we examined whether Gtf2ird1 could bind to the promoter region of *Ttr* to regulate the expression of *Ttr*. Chromatin immunoprecipitation followed by qPCR results with eNPCs showed that Gtf2ird1 indeed binds to the promoter region of Ttr (**Figure 9M**). Collectively, these results suggested that *Gtf2ird1* deficiency leads to abnormal neurogenesis through regulating Ttr in mice.

### Discussion

In the present study we established a forebrain organoid model and identified aberrant neurogenesis deficits in WS organoids (**Figure 9—figure supplement 2**). Bulk RNA-seq revealed the altered gene expression related to neuronal development. scRNA-seq revealed an altered developmental trajectory and aberrant excitatory neurogenesis in WS organoids. Mechanistically, we showed that transthyretin (TTR) had a significantly decreased expression in WS organoids. GTF2IRD1 encoded by one WS gene *GTF2IRD1* regulated the expression of TTR. Furthermore, exogenous TTR could rescue the neurogenesis deficits of WS organoids. Collectively, our data highlight the essential function of GTF2IRD1-TTR axis for mammalian neuronal development and provide novel insights into mechanisms underlying the abnormal neurogenesis in WS brain.

Individuals with WS clinically exhibited diverse neuronal deficits including hypersociability, autistic deficits, impaired myelination, intellectual disability and abnormal brain structure due to the altered

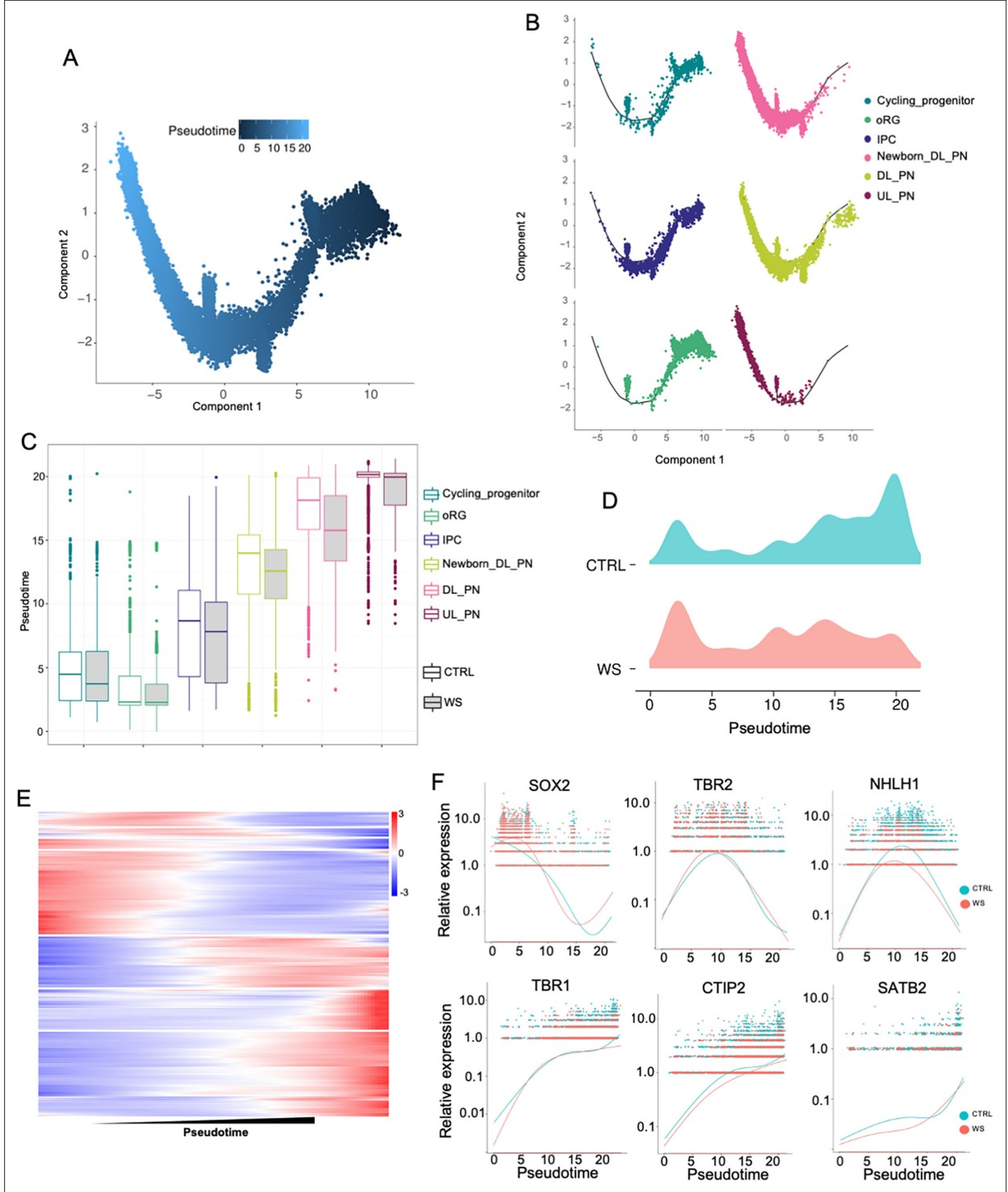

**Figure 5.** WS forebrain organoids exhibit aberrant developmental trajectory of excitatory neurons. (**A**) Pseudo-time distribution of cell trajectories of excitatory neurons related cell clusters. (**B**) Distribution of cell cluster in Pseudotime trajectories. (**C**) Boxplot showing the distribution of pseudotime within each cell type in both CTRL and WS-derived forebrain organoids. (**D**) Mountain maps of pseudotime uniform manifold approximation in both CTRL and WS forebrain organoids. (**E**) Heatmap of pseudotime line-dependent gene expression. (**F**) Expression of *SOX2,TBR2, NHLH1, TBR1, CTIP2* and *SATB2* ordered by Monocle analysis in pseudotime.

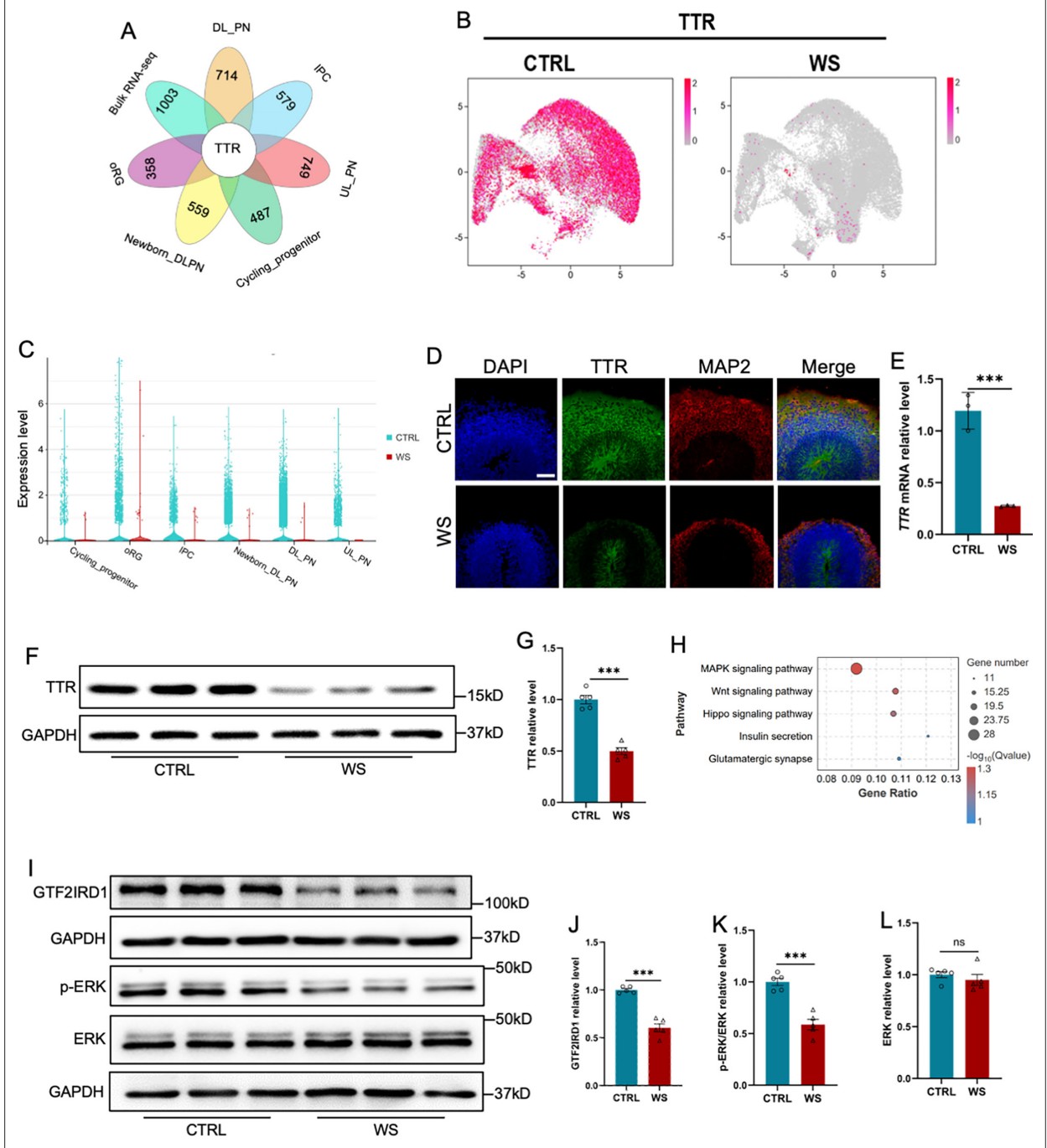

**Figure 6.** The deficiency of *GTF2IRD1* leads to the down-regualtion of *Transthyretin* and inhibition of ERK signaling. (**A**) Integrated analysis of down-regualted genes in 6 clusters relating to the generation of excitatory neurons and down-regualted genes revealed by bulk RNA-seq. (**B**) UMAP visualization of the expression of *TTR* in CTRL and WS forebrain organoids, respectively. (**C**) *Transthyretin* (TTR) displayed the decreased expression in 6 clusters relating to the generation of excitatory neurons. (**D**) Representative images of TTR immunofluorescence staining with in CTRL and WS forebrain organoids, respectively. Scale bar, 100 μm. (**E**) qRT-PCR results showed that *TTR* level significantly decreased in WS forebrain organoids compared to Ctrl group. Values represent mean ± SEM; n=3 independent experiments; *p<0.05, **p<0.01, ***p<0.001; unpaired Student's t test. (**F, G**) WB assay (**F**) and quantification results (**G**) showed that the level of TTR was significantly decreased in WS brain organoids compared to Ctrl. Values represent mean ± SEM; n=5 independent experiments; *p<0.05, **p<0.01, ***p<0.001; unpaired Student's t test. (**H**) KEGG analysis showed that down-regualted genes enriched for MAPK pathway, etc. (**I–K**) WB assay (**I**) and quantification results showed that the level of phosphorylated ERK (p-ERK) was significantly decreased (**J**), but the level of total ERK (**K**) was not changed in WS brain organoids compared to Ctrl. Values represent mean ± SEM; n=5 independent experiments; *p<0.05, **p<0.01, ***p<0.001; unpaired Student's t test. (**L, M**) WB assay (**L**) and quantification results (**M**) showed that the level of

*Figure 6 continued on next page*

*Figure 6 continued*

GTF2IRD1 was significantly decreased in WS brain organoids compared to Ctrl. Values represent mean ± SEM; n=5 independent experiments; *p<0.05, **p<0.01, ***<0.001; unpaired Student's t test.

The online version of this article includes the following source data and figure supplement(s) for figure 6:

**Source data 1.** PDF file for original blots images in *Figure 6F and I*.

**Source data 2.** TIF files for original blots images in *Figure 6F and I*.

**Figure supplement 1.** The level of TTR decreases in WS organoids.

neuronal development (*Kozel et al., 2021*; *Osborne, 2010*; *Pober, 2010*). Distinct models have been applied to investigate the mechanisms underlying these defects (*Alesi et al., 2021*; *Barak and Feng, 2016*; *Barak et al., 2019*; *Chailangkarn et al., 2016*; *Corley et al., 2016*; *Dai et al., 2009*; *Enkhmandakh et al., 2009*; *Kopp et al., 2019*; *Kopp et al., 2020*; *Kozel et al., 2021*; *Mervis et al., 2012*; *Morris and Braddock, 2020*; *Osborne, 2010*; *Tassabehji et al., 2005*). Brain organoids derived iPSCs under 3D culturing condition have been providing a unique model to investigate human brain development and neurological diseases (*Amin and Paşca, 2018*; *Benito-Kwiecinski and Lancaster, 2020*; *Clevers, 2016*; *Di Lullo and Kriegstein, 2017*; *Kelava and Lancaster, 2016*; *Lancaster et al., 2013*; *Li and Izpisua Belmonte, 2019*; *Paşca, 2019*; *Trujillo and Muotri, 2018*). Utilizing the forebrain organoids together with scRNA-seq, our study has revealed the aberrant capabilities of proliferation and differentiation of WS NPCs. Of note, our study revealed for the first time an altered developmental trajectory and a remarkably reduced excitatory neurogenesis in WS organoids. Therefore, our findings provide a new insight regarding the deficits of neurogenesis and neuronal development in WS brain.

Genotype-phenotype relationship analysis suggests that among 25–27 WS genes, each gene could be responsible for distinct phenotypes and could contribute to WS phenotype at varying degrees (*Kozel et al., 2021*; *Lek et al., 2016*; *Tassabehji, 2003*). For example, LIMK1, CYLN2, DSCAM, PAK1, GTF2I and BAZ1B contributed for neurological phenotypes (*Barak et al., 2019*; *Lalli et al., 2016*; *Pinelli et al., 2020*; *Todorovski et al., 2015*; *van Hagen et al., 2007*), ELN for blood vessel development (*Li et al., 1998*), and DNAJC30 for mitochondrial function (*Tebbenkamp et al., 2018*). As a member of the transcription factor II (TF-II), GTF2IRD1 regulates gene expression and involves multiple processes, such as embryonic development, cell cycle and neuronal development (*Enkhmandakh et al., 2009*; *Kopp et al., 2020*; *O'Leary and Osborne, 2011*; *Palmer et al., 2007*; *Tassabehji et al., 2005*; *Thompson et al., 2007*; *van Hagen et al., 2007*). Our findings reveal a wide expression of *GTF2IRD1* in all identified cell clusters. Intriguingly, the deletion of *GTF2IRD1*, but not GTF2I, caused a remarkable decrease of *transthyretin* (*TTR*). TTR is widely expressed in the neuronal system and promotes neuronal development (*Buxbaum et al., 2008*; *Gomes et al., 2016*; *Li et al., 2011*; *Magalhães et al., 2021*). TTR deficient mice displayed a reduced neurogenesis and increased oligodendrogenesis (*Alshehri et al., 2020*; *Vancamp et al., 2019*). Our results showed that *Gtf2ird1* deficiency also induces abnormal neurogenesis and neurodevelopmental deficits and reduces TTR in mice. Consistently, TTR had decreased expression in the postmortem frontal cortex of WS patients (*Barak et al., 2019*). Our results also show that TTR can rescue neuronal deficits of NPCs in WS organoids through activating ERK signaling. Therefore, our findings reveal a novel mechanism underlying the aberrant neurogenesis and neuronal development in WS brain.

In summary, our study has showed the abnormal neurogenesis and neuronal differentiation, especially the aberrant excitatory neurogenesis in WS organoids. Our study has revealed that the dysfunction of GTF2IRD1-TTR axis plays essential roles for the neurodevelopmental deficits of WS brain. Our findings also suggest that TTR could be a potential target for the treatment of WS at clinic.

## Methods

### Animals

*Gtf2ird1*[+/-] mice were purchased from Model Animal Research Center (Nanjing, China). *Gtf2ird1*[+/-] male mice were crossed with *Gtf2ird1*[+/-] female mice to generate *Gtf2ird1*[+/+] (wild-type), *Gtf2ird1*[+/-] (heterozygous, Het) and *Gtf2ird1*[-/-] (homozygous, KO) mice. Animals were in C57BL/6 genetic background and housed in the animal center of Zhejiang University under 12 hour light/12 hour dark conditions

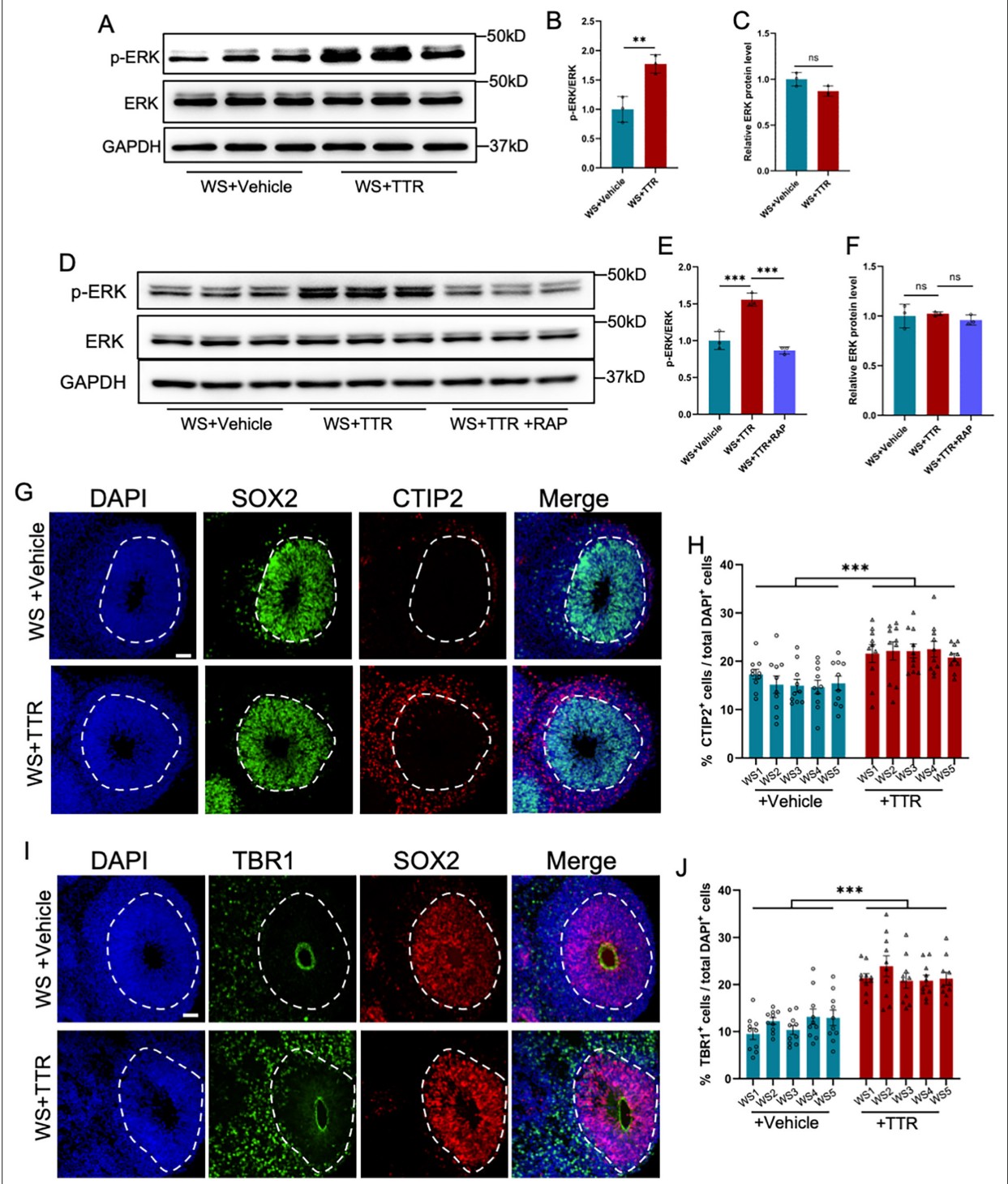

**Figure 7.** Recombinant TTR rescues neurodevelopmental defects of WS organoids. (**A–C**) WB assay (**A**) and quantification results showed that TTR exposure enhanced the level of p-ERK in WS brain organoids (**B**), while the level of total ERK was not affected (**C**). Values represent mean ± SEM; n=3; *p<0.05, **p<0.01, ***p<0.001; unpaired Student's t test. (**D–F**) Western blot (**D**) and quantification results showed that TTR exposure enhanced the level of p-ERK (**E**), which could be inhibited by receptor-associated protein (RAP) in WS brain organoids (**F**). Values represent mean ± SEM; n=3 independent experiments; *p<0.05, **p<0.01, ***p<0.001; unpaired Student's t test. (**G**) Representative images of SOX2 and CTIP2 immunofluorescence staining with WS-derived forebrain organoids under different culture conditions. Scale bar, 50 μm. (**H**) Quantification results showed that TTR exposure significantly increased the proportion of CTIP2+ neurons in WS-derived forebrain organoids compared to CTRL. Values represent mean ± SEM; n=30–35 VZ-like regions of 3–5 organoids from three WS patients in each group. *p<0.05, **p<0.01, ***p<0.001; unpaired Student's t test. (**I**) Representative

*Figure 7 continued on next page*

*Figure 7 continued*

images of SOX2 and TBR1 immunofluorescence staining with WS-derived forebrain organoids under different culture conditions. Scale bar, 50 µm. (**J**) Quantification results showed that TTR exposure significantly increased the proportion of TBR1+ neurons in WS-derived forebrain organoids. Values represent mean ± SEM; n=30–35 VZ-like regions of 3–5 organoids from three WS patients in each group. *p<0.05, **p<0.01, ***p<0.001; unpaired Student's t test.

The online version of this article includes the following source data and figure supplement(s) for figure 7:

**Source data 1.** PDF file for original blots images in *Figure 7A and D*.

**Source data 2.** TIF files for original blots images in *Figure 7A and D*.

**Figure supplement 1.** Schematic illustration of exogenous TTR application.

with free access to food and water. The genotypes of animals were determined with PCR. The used primers included: 5'TGGAAGCCCAGTGACTACTTG and 5'ACATAGGCATAAGGGCCCAG (for wild-type band), and 5'CAATGAGAGCGTCTTCGTGAT and 5'ACCATTGAAAAGTAGAGTAGAATGC (for knockout band). All animal experiments were performed following with the protocols approved by the Zhejiang University Animal Care and Ethics Committee.

## The establishment and culture of iPSCs

Peripheral blood samples of three male patients with WS and three healthy males were were collected at the Children's Hospital, School of Medicine, Zhejiang University following the protocol approved by the ethics committee of Zhejiang university Medical University Children's hospital ([2016]NO.326). After the isolation of peripheral blood mononuclear cells (PBMCs) and urinary cells, respectively, the reprogramming was carried out following the manufactuer's protocol (Thermo Fisher, Cat# A16517). Briefly, around 1X10$^5$ PBMCs were transduced using the Sendai reprogramming vectors at appropriate MOI in 24-well plates. For feeder-free culture, iPSCs were cultured with mTeSR 1 medium (STEMCELL) on Matrigel (Corning). For feeder-dependent culture, iPSCs were cultured with DMEM/F12 medium (Corning) consisting of 20% KnockOut Serum Replacement (Gibco), 1 X GlutaMAX (Gibco), 1 X MEM Non-Essential Amino Acids (Gibco), 100 µM 2-Mercaptoethanol (Sigma), 10 ng/mL human FGF-basic (PeproTech) on the inactivated mouse embryonic fibroblasts (MEF). The information of used reagents could be found in *Supplementary file 2*.

## The culture of forebrain organoids

The generation and maintenance of forebrain organoids were carried out as described previously with moderation (*Kang et al., 2021a*; *Qian et al., 2016*). iPSCs colonies on MEF were digested with 1 mg/mL Collagenase Type IV (Gibco) at 37 °C for 1 hour. After the digestion, the colonies were resuspended with the forebrain first medium (DMEM/F12, 20% KnockOut Serum Replacement, 1 x GlutaMAX, 1 x MEM Non-Essential Amino Acids, 100 µM 2-Mercaptoethanol, 2 µM A83-01, 2 µM Dorsomorphin) and cultured in low-attachment six-well plate (Corning) for 4 days. On day 5 and day 6, the medium was halfly replaced with forebrain second medium containing DMEM/F12 (Corning), 1 X GlutaMAX (Gibco), 1 X MEM Non-Essential Amino Acids (Gibco), 1X N2 (Gibco), 1 µM SB-431542 (STEMCELL), 1 µM CHIR99021 (STEMCELL), 4 ng/mL WNT-3A (R&D Systems), 10 µg/mL Heparin (Sigma). On day 7, EBs were embedded with Matrigel using forebrain second medium in order to induce the neuroepithelium. On day 14, organoids were mechanically separated and transferred to 12-well spin Ω bioreactor for culture, and the medium was changed to forebrain third medium, consisting of DMEM/F-12 (Corning), 1 X GlutaMAX (Gibco), 1 X MEM Non-Essential Amino Acids (Gibco),1X N2 (Gibco), 1XB27 (Gibco), 2.5 µg/mL insulin (Sigma), 1 X P/S (Gibco),100µM 2-Mercaptoethanol (Sigma). From day 35 to day 56, 1% Matrigel was added to the forebrain third medium to remodel the extracellular matrix (ECM) in 12-well spinning bioreactors.

## Targeted panel sequencing

DNA was extracted from the peripheral blood of healthy control and WS patients, and purified with GeneJET Whole Blood Genomic DNA Purification Mini Kit (Thermo Fisher Scientific). DNA purity was identified by NanoDrop spectrophotometers (Thermo Fisher, MA, USA). DNA concentration was quantified by Qubit DNA Assay Kit in Qubit 3.0 Fluorometer (Life Technologies, CA). For each sample, a total of 1 µg of genomic DNA was used for library preparation with the NimbleGen SeqCap

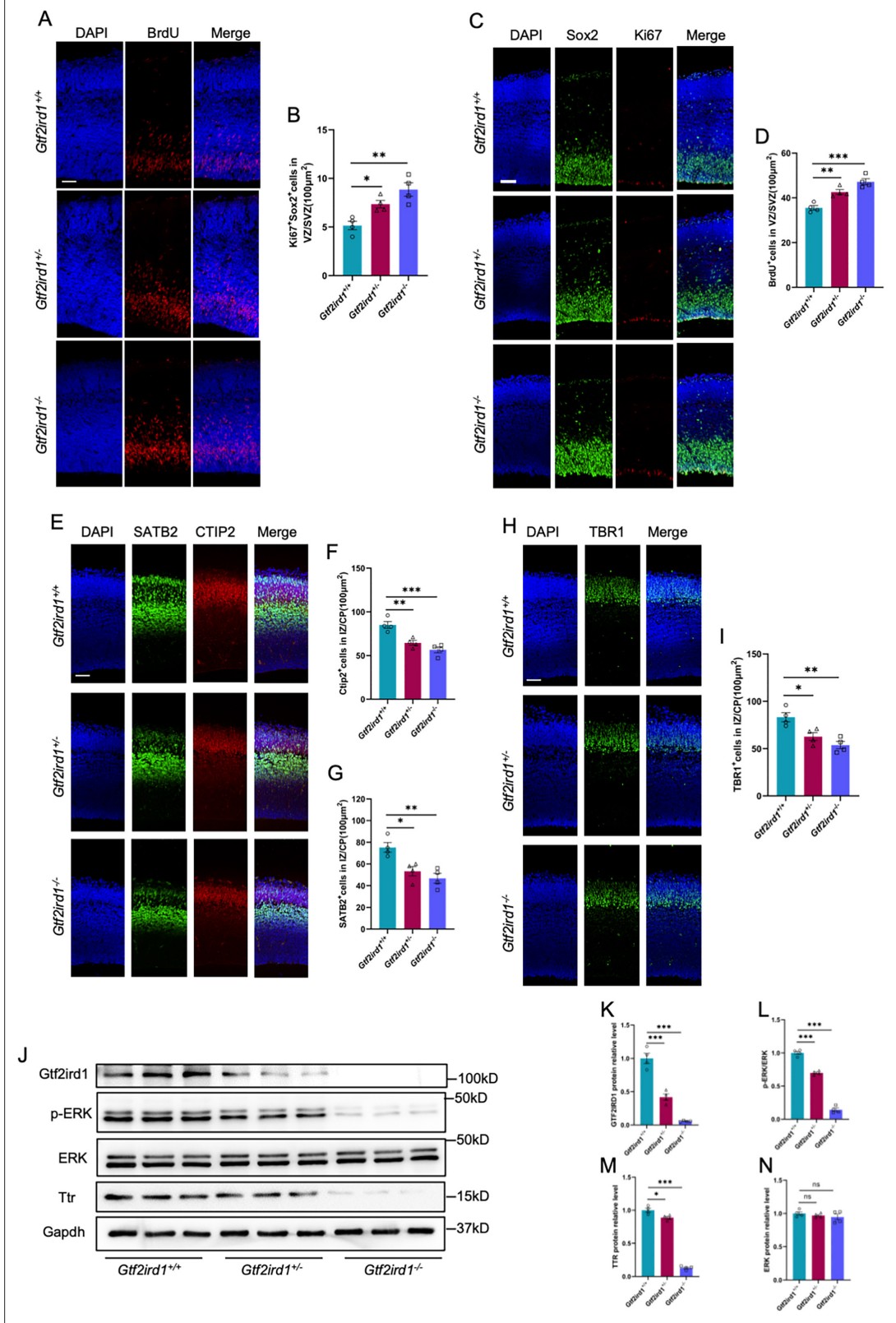

**Figure 8.** *Gtf2ird1* deficient mice display neurodevelopmental deficits and the reduced TTR. (**A, B**) Representative images of BrdU immunofluorescence staining (**A**) and quantification results (**B**) showed the increased BrdU⁺ eNPCs in VZ/SVZ of *Gtf2ird1⁺/⁻* (heterozygous, Het) and *Gtf2ird1⁻/⁻* (knockout, KO) mice compared to *Gtf2ird1⁺/⁺* (Wild-type, WT) mice, respectively. Pups were injected with BrdU at E15.5 and sacrificed 1.5 hr later. Values represent mean ± SEM; n=4 animals for each group; *p<0.05, **p<0.01, ***p<0.001; unpaired Student's t test. Scale bar, 50 μm. (**C, D**) Representative images of

*Figure 8 continued on next page*

*Figure 8 continued*

Sox2 and Ki67 immunofluorescence staining (**C**) and quantification results (**D**) showed the increased Ki67$^+$Sox2$^+$ eNPCs in VZ/SVZ of Het and KO mice compared to WT mice, respectively. Values represent mean ± SEM; n=4 animals for each group; *p<0.05, **p<0.01, ***p<0.001; unpaired Student's t test. Scale bar, 50 μm. (**E–G**) Representative images of neuronal marker SATB2 and CTIP2 immunofluorescence staining (**E**) and quantification results showed the decreased Ctip2$^+$ (**F**) and SATB2$^+$ (**G**) neurons of Het and KO mice compared to WT mice, respectively. Values represent mean ± SEM; n=4 animals for each group; *p<0.05, **p<0.01, ***p<0.001; unpaired Student's t test. Scale bar, 50 μm. (**H, I**) Representative images of neuronal marker Tbr1 immunofluorescence staining (**H**) and quantification results (**I**) showed the decreased Tbr1$^+$ neurons in IZ/CP of Het and KO mice compared to WT mice, respectively. Values represent mean ± SEM; n=4 animals for each group; *p<0.05, **p<0.01, ***p<0.001; unpaired Student's t test. Scale bar, 50 μm. (**J–N**) WB assay (**J**) and quantification results showed that the levels of Gtf2ird1 (**K**), p-ERK (**L**) and TTR (**M**) were significantly decreased, while the level of total ERK (**N**) was not altered in the cortical tissues of Het and KO mice compared to WT mice, respectively. Values represent mean ± SEM; n=4 animals for each group; *p<0.05, **p<0.01, ***p<0.001; unpaired Student's t test.

The online version of this article includes the following source data for figure 8:

**Source data 1.** PDF file for original blots images in *Figure 8J*.

**Source data 2.** TIF files for original blots images in *Figure 8J*.

EZ Human Exome V3 following manufacturer's instructions (Basel, Swiss). Library concentration was measured by QubitDNA Assay Kit in Qubit 3.0, and insert size was detected by the Agilent B ioana-lyzer 2100 system (Agilent Technologies, CA, USA). The clustering of the index coded samples was performed on a cBot Cluster Generation System according to the manufacturer's recommendations (Illumia, USA). After cluster generation, the library preparations were sequenced on an Illumina Hiseq Ten platform with 150 bp paired end module. The panel kit was customized from Agilent Technologies, Design ID: 3249861, Cat#5190–4816, Species: *H. sapiens* (UCSC hg19, GRCh37).

## Total RNA isolation and quantitative real-time PCR

Total RNA was extracted with TRIzol reagent following the manufacturer's protocol (Thermo Fisher Scientific). The concentration of RNA was quantified using a NanoDrop spectrophotometer 2000 (Thermo Fisher Scientific). 0.5 mg of total RNA was used for reverse transcription, and standard real-time qPCR assays were performed using SYBR Green (Vazyme) in triplicates. The results were analyzed using the $^{\triangle\triangle}$Ct method. The information of used primers could be found in *Supplementary file 3*.

## Bulk RNA-seq and data analysis

RNA quality was determined using Agilent 2100 Bioanalyzer. After enriched by Oligo(dT) beads, mRNA was fragmented into short fragments using fragmentation buffer and reverse transcribed into cDNA using NEBNext Ultra RNA Library Prep Kit for Illumina (NEB, USA). Purified double-stranded cDNA fragments were end-repaired, base added and ligated to Illumina sequencing adapters. The ligation reaction was purified with the AMPure XP Beads(1.0 X). Ligated fragments were subjected to size selection by agarose gel electrophoresis followed by PCR amplification. Library quality was assessed on the Agilent Bioanalyzer 2100 system (Agilent Technologies Inc) and sequenced on an Illumina Hiseq platform (Illumina NovaSeq6000).

To obtain high-quality clean reads, raw data of fastq format were processed by removing reads containing adapter, reads containing ploy-N and low-quality reads from raw data. FPKM (Fragments Per Kilobase of transcript sequence per Millions base pairs sequenced) of each transcript was calculated based on the length of the gene and reads count mapped to this gene. Transcripts with the parameter of false discovery rate (FDR) below 0.05 and absolute fold change ≥2 were considered differentially expressed transcripts.

## Single cell RNA-sequencing (scRNA-seq)

Three pairs of Day 56 forebrain organoids derived from iPSCs of three CTRL and WS patients were dissociated with 3 mL Tryp-LE (Sigma-Aldrich) for 15 min at 37 °C, respectively. A single-cell suspension was subsequentialy collected at a cell density of 1000 cells/μL, and were added to a 10 X Genomics GemCode Single-cell instrument to generates a single-cell Gel Bead-In-EMlusion (GEMs). The leftover biochemical reagents and primers were removed with silane magnetic beads. Full-length, barcoded cDNAs were then amplified by PCR to generate sufficient amount of DNA for library construction. Libraries were sequenced by Chromium Next GEM Single Cell 3' Reagent Kits v3.1. Single Cell 5' 16 bp 10 x Barcode and 10 bp UMI were encoded in Read 1, while Read 2 was used to sequence

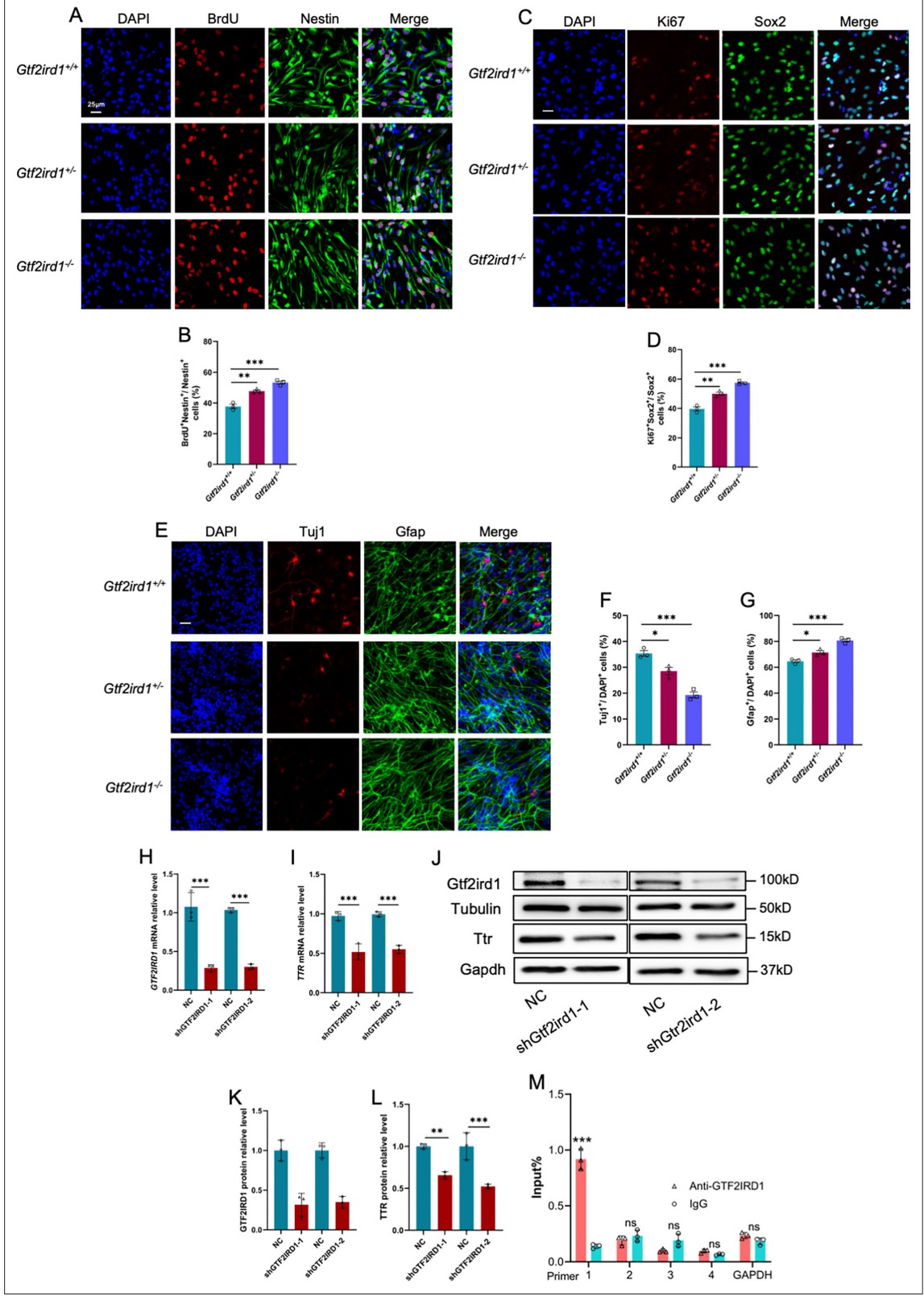

**Figure 9.** Gtf2ird1 regulates the proliferation and differentiation of eNPCs in vitro. (**A**) Representative images of BrdU and Nestin immunofluorescence staining with cultured eNPCs from WT, Het and KO mice, respectively. Scale bar, 25 μm. (**B**) Quantification results showed that the percentage of BrdU+/BrdU+Nestin+ eNPCs was significantly increased in Het and KO mice relative to WT cells, respectively. Values represent mean ± SEM; n=3 independent experiments; *p<0.05, **p<0.01, ***p<0.001; unpaired Student's t test. (**C**) Representative images of Ki67 and Sox2 immunofluorescence

*Figure 9 continued on next page*

*Figure 9 continued*

staining with cultured eNPCs from WT, Het and KO mice, respectively. Scale bar, 25 μm. (**D**) Quantification results showed that the percentage of Ki67[+]/Ki67[+]Sox2[+] eNPCs was significantly increased in Het and KO mice relative to WT cells, respectively. Values represent mean ± SEM; n=3 independent experiments; *p<0.05, **p<0.01, ***p<0.001; unpaired Student's t test. (**E**) Representative images of Tuj1 and Gfap immunofluorescence staining with cultured eNPCs from WT, Het, and KO mice, respectively. Scale bar, 25 μm. (**F, G**) Quantification results showed that the nubmer of Tuj1[+] neurons was significantly decreased (**F**), but the number of Gfap +astrocytes was significantly increased (**G**) upon the differentiation of eNPCs in Het and KO mice relative to WT cells, respectively. Values represent mean ± SEM; n=3 independent experiments; *p<0.05, **p<0.01, ***p<0.001; unpaired Student's t test. (**H, I**) qRT-PCR results showed that acute knockdown of *Gtf2ird1* significantly reduced the level of *Gtf2ird1* (**H**) and *TTR* (**I**) in WT eNPCs compared to scramble group (NC), respectively. Values represent mean ± SEM; n=3 independent experiments; *p<0.05, **p<0.01, ***p<0.001; unpaired Student's t test. (**J–L**) WB assay (**J**) and quantification results showed that acute knockdown of *Gtf2ird1* significantly reduced the level of Gtf2ird1 (**K**) and TTR (**L**) in WT eNPCs, respectively. Values represent mean ± SEM; n=3 independent experiments; *p<0.05, **p<0.01, ***p<0.001; unpaired Student's t test. (**M**) Chromatin immunoprecipitation followed qPCR (ChIP-qPCR) showed that Gtf2ird1 binds to the promoter regions of TTR. Values represent mean ± SEM; n=3 independent experiments; *p<0.05, **p<0.01, ***p<0.001; unpaired Student's t test.

The online version of this article includes the following source data and figure supplement(s) for figure 9:

**Source data 1.** PDF file for original blots images in *Figure 9J*.

**Source data 2.** TIF files for original blots images in *Figure 9J*.

**Figure supplement 1.** *Gtf2ird1* deficiency induces the abnormal proliferation and differentiation of embryonic neuronal progenitor cells in vitro.

**Figure supplement 2.** Schematic diagram illustrating the roles of GTF2IRD1-TTR-ERK pathway in regulating neurogenesis and neurodevelopment.

cDNA fragment. Sample index sequences were incorporated as the i7 index read. Read 1 and Read 2 were standard Illumina sequencing primer sites used in paired-end sequencing.

## scRNA-seq data analysis

10 X Genomics Cell Ranger software (version 3.1.0) was used for the conversion of raw BCL files to FASTQ files, alignment and counts quantification. Reads with low-quality barcodes and UMIs were filtered out and then mapped to the reference genome (human GRCh38). Reads were uniquely mapped to the transcriptome and intersecting an exon at least 50% were considered for UMI counting. Before quantification, the UMI sequences would be corrected for sequencing errors, and valid barcodes were identified based on the Empty Drops method (*Lun et al., 2019*). The cell by gene matrices were produced via UMI counting and cell barcodes calling. The cell by gene matrices for each sample were individually imported to Seurat version 3.1.1 for downstream analysis (*Butler et al., 2018*). Cells with unusually high number of UMIs 8000 or mitochondrial gene percent 10% were filtered out. We also excluded cells with less than 500 or more than 4000 genes detected. Expression value of each gene in given cluster were compared against the rest of cells using Wilcoxon rank sum test. Cell clusters were annotated to known biological cell types using canonical cell marker genes (*Paulsen et al., 2022*). Significant upregulated genes were determined based on (1) at least 1.28-fold increased in the target cluster; (2) expressed in more than 25% of the cells in the target cluster; (3) p value is less than 0.05.

Gene ontology (GO) analysis was performed using the DAVID database as described previously (*Chen et al., 2021*). Each enriched GO function term is represented by a node and the node size is proportional to the number of genes in its corresponding function term in the enrichment maps. Similar GO functions are categorized as one cluster. The function term and the number of genes in each cluster are labelled.

## Isolation and culture of embryonic mouse NPCs

Embryonic mouse neural progenitor cells (eNPCs) were isolated and cultured as described previously (*Li et al., 2017*, *Li et al., 2021*). Briefly, the cerebral cortex of mice (Embryonic day 14, Day 14) was dissected and treated with 0.1% Trypsin for 5 min in 37 °C. Then cells were cultured with DMEM/F-12 medium with 2% B27 (minus vitamin A), 1% antibiotic-antimycotic, 20 ng/mL FGF-2, 20 ng/mL EGF, and in a 5% $CO_2$ incubator at 37 °C. eNPCs were maintained in proliferating conditions for 2 weeks. For the differentiation assay, eNPCs were cultured with DMEM/F-12 medium containing 2% B27 (minus vitamin A), 2 mM L-glutamine, 1% antibiotic-antimycotic,1 μM retinoic acid and 5 μM forskolin (differentiation medium) for 48 hr. For BrdU incorporation assay, BrdU was supplemented at the final concentration of 5 μM for 8 hr. At the scheduled time point, eNPCs were fixed with 4% paraformaldehyde (PFA) followed by immunostaining with proper antibodies.

## Culture of HEK293T cells and plasmid transfection

HEK293T cells were cultured with DMEM medium containing 10% FBS, 2 mM L-glutamine, and 1% antibiotic-antimycotic in a 5% $CO_2$ incubator at 37 °C, and medium was replaced every day. After the cell density reaches about 70%, half volume of medium was changed with fresh medium 6 hr before the transfection. 4 μg of plasmids and 6 μl of Lipo2000 were mixed with 100 μl of Opti-MEM at room temperature for 10 min, respectively. Reagents were mixed together and further reacted at room temperature for 10 min. The mixture was added to one well of a six-well plate. The medium was replaced with fresh medium 8–10 hr after transfection. Cells were harvested for assays 48 hrs post-transfection.

## Immunofluorescence staining

Organoids were collected and fixed with 4% paraformaldehyde for 4 hr. After washed with 1 X PBS and dehydrated with 30% sucrose solution at 4 °C, brain organoids were embedded with O.C.T (SAKURA) and sections of 10 μm thickness were prepared with a cryostat (Leica).

Embryonic day 15.5 (E15.5) pups were hypothermic anesthetized with ice, and transcardially perfused with cold phosphate buffer saline (PBS) followed by cold 4% paraformaldehyde (PFA). Brain samples were removed, and post-fixed with 4% PFA overnight and completely dehydrated with 30% sucrose at 4 °C. Brain samples were embedded with OCT (4583, SAKURA) and 30 μm thickness sections were prepared with a cryostat (CM1950, Leica). For BrdU assay, E15.5 pups were injected with BrdU (0.1 mg/10 g dody weight) and sacrificed 1.5 hr later.

To perform immunofluorescence staining, section samples were incubated with blocking buffer for 1 hr and applied with proper primary antibodies at 4 °C overnight. The next day, samples were washed with 1 x PBS followed by the application of proper secondary antibodies. Images were taken with a confocal microscope (Olympus, FV3000). The quantification of the thickness and cell number of were performed with imageJ software as described previously (*Kang et al., 2021b*; *Qian et al., 2016*). For stratification and thickness analysis of brain organoid, ventricular zone (VZ) layers were defined as SOX2$^+$ and CTIP2$^-$ or TBR1$^-$ (MAP2$^-$) regions. The cortical plate (CP) layer is defined as CTIP2$^+$ or TBR1$^+$ or MAP2$^+$ regions.

## Click-iT EdU labeling

Organoids were applied with 10 μM EdU for 2 hr or 24 hr, respectively. At the scheduled timepoint, cells were washed with 1 X PBS, and subsequentially fixed with 4% paraformaldehyde and dehydrated with 30% sucrose. Brain organoid sections were prepared with a cryostat (Leica). EdU staining was performed following the manufactuer's mannual (Invitrogen). Primary and secondary antibodies were applied. Images were taken with a confocal microscope (Olympus, FV3000).

## Plasmid transfection and lentivirus packaging

Scramble shRNA (5'-TTCTCCGAACGTGTCACGT-3'), shRNAs targeting human *GTF2I* (5'-CGGATGAG TGTAGATGCTGTA –3'), human *GTF2IRD1* (5'- GCGCCATAGTATCCACTTCAT-3') and mouse *Gtf2ird1* (#1, 5'-GTGCCCTACAAGAGAATCAAA-3'; #2, 5'-GTGAAGCTCTGGGCATCAAAT-3') were cloned into lentivirus vector, respectively. Lentiviruses were packaged, and used to infect HEK293T cells and eNPCs. 48 hr later, the cells were collected or fixed with 4% paraformaldehyde for the following experiments.

## Western blot

The collected organoids and cell samples were treated with RIPA buffer (Abcam) containing 1 X protease inhibitor cocktail (MCE). Protein concentration was measured with a Biophotometer (Eppendorf), and after denaturation, 20–40 μg proteins were subjected to SDS-polyacrylamide gel electrophoresis. Signals were detected with Tanon Detection system (Tanon) and the intensities of the immunoblot bands were normalized to Gapdh or Tubulin. The information of primary and secondary antibodies used can be found in *Supplementary file 2*. Western blot images were analyzed using ImageJ software.

## TTR exposure

WS brain organoids at day 28 were exposed to TTR recombinant protein (55 μg/ml) and collected for assays at day 56 followed by immunostaining. Some organoids were also treated with TTR only, TTR

plus RAP (one TTR inhibitor; 55 µg/ml) and collected for assays at day 56 followed by western blot assays.

## Chromatin immunoprecipitation-qPCR assay

Chromatin immunoprecipatation (ChIP) experiment was performed following the manufactuer's protocol (Beyotime Biotechnology, China). Briefly, 250 µl of 37% formaldehyde was added to cultured cells for cross-link for 10 min at 37 °C. The reaction was then immediately terminated with 125 mM glycine solution for 5 min. Then cells were washed with PBS containing 1 mM and resuspended in 200 µL SDS lysis buffer containing 1 mM PMSF in an ice bath for 10 min. Samples were sonicated into DNA fragments of 200–600 bp size using a sonicator. 20 µl sample was kept as input control, and the left was incubated with anti-IgG (Proteintech) and anti-GTF2IRD1 antibody (RD system) overnight at 4 °C, respectively. The next day, after centrifugation of sample, pellets was washed with Low Salt Immune Complex Wash Buffer, High Salt Immune Complex Wash Buffer, LiCl Immune Complex Wash Buffer and TE Buffer. Samples were de-crosslinked and DNA was purified by a PCR purification kit according to the manual (Beyotime biotechnology, China). qPCR reaction was performed to measure enrichment with 4 pairs of primers designed within the promoter region of *TTR*, and *GAPDH* was used as a internal control. Fold enrichment was calculated by the comparative threshold cycling method using Equation $2^{-\Delta\Delta ct}$. Primer sequences were provided in ***Supplementary file 3***.

## Statistical analysis

Data were presented as mean ± SEM. Statistical analysis between group differences was performed with unpaired Student's t test or one way ANOVA using Graph Prism software (version 9.0, GraphPad). For multiple group comparisons, a one-way ANOVA followed by Dunnett's multiple-comparison test. p value less than 0.05 was considered as statistical significance. Replicate information is indicated in the figure legends.

## Acknowledgements

This work was supported by the National Natural Science Foundation of China (82371182 to XL) and the National Key Research and Development Program of China (2017YFE0196600 to XL), and Central Guiding Fund for Local Science and Technology Development Projects (2023ZY1058 to QS).

## Additional information

### Funding

| Funder | Grant reference number | Author |
|---|---|---|
| National Natural Science Foundation of China | 82371182 | Xuekun Li |
| Key Research and Development Program of Zhejiang Province | 2023C03018 | Xuekun Li |
| Central Guiding Fund for Local Science and Technology Development Projects | 2023ZY1058 | Qiang Shu |

The funders had no role in study design, data collection and interpretation, or the decision to submit the work for publication.

### Author contributions

Xingsen Zhao, Formal analysis, Investigation; Qihang Sun, Yikai Shou, Weijun Chen, Mengxuan Wang, Xiaoli Huang, Ying Li, Chao Wang, Yan Gu, Investigation; Wenzheng Qu, Formal analysis; Chai Ji, Investigation, Conceptualization; Qiang Shu, Conceptualization, Supervision, Funding acquisition; Xuekun Li, Conceptualization, Formal analysis, Supervision, Funding acquisition, Writing - original draft, Project administration, Writing - review and editing

## Author ORCIDs
Xingsen Zhao (ID) https://orcid.org/0000-0003-1721-0859
Yan Gu (ID) https://orcid.org/0000-0001-5562-6279
Xuekun Li (ID) https://orcid.org/0000-0002-6985-6363

## Ethics

This study was approved by the ethics committee of Zhejiang University Medical University Children's Hospital ([2016]NO.326). Before the initiation of the study, investigators clearly explained to patients and their guardians about the aim of study and 2-3 ml blood samples were to be collected. A formal informed consent was obtained from the patients' guardians.

This study was approved by the ethics committee of Zhejiang University Medical University Children's Hospital ([2016]NO.326).

Reviewer #1 (Public Review): https://doi.org/10.7554/eLife.98081.2.sa1
Reviewer #2 (Public Review): https://doi.org/10.7554/eLife.98081.2.sa2

# Additional files

## Supplementary files

- Supplementary file 1. DEGs in WS brain organoids.
- Supplementary file 2. Information of reagents.
- Supplementary file 3. List of primers.
- MDAR checklist

## Data availability

Bulk RNAs-seq and scRNAs-seq data have been deposited in GEO: GSE283473 and GSE283474.

The following datasets were generated:

| Author(s) | Year | Dataset title | Dataset URL | Database and Identifier |
| --- | --- | --- | --- | --- |
| Zhao X, Li X | 2024 | Gene expression profile at single cell level of brain organoid at day56 of ctrl and WS | https://www.ncbi.nlm.nih.gov/geo/query/acc.cgi?acc=GSE283473 | NCBI Gene Expression Omnibus, GSE283473 |
| Zhao X, Li X | 2024 | Gene expression profile level of mRNA of brain organoid at day56 of ctrl and WS | https://www.ncbi.nlm.nih.gov/geo/query/acc.cgi?acc=GSE283474 | NCBI Gene Expression Omnibus, GSE283474 |

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
