## [Editor Report · eLife assessment]

Zhao et al. report **valuable** adverse effects on cell proliferation, differentiation and gene expression, possibly linked to reduced binding activity of the transcription factor GTF2IRD1 to the transthyretin (TTR) promoter, in a human forebrain organoid model of Williams Syndrome (WS). The authors provide **incomplete** evidence of the effects of GTF2IRD1, a mutated gene in WS, on altering MAPK/ERK pathway activity, a well-recognized target in cell proliferation.

---

## [Referee Report · Reviewer #1 (Public Review)]

Summary:

Zhao et al. used the human forebrain organoid model, transgenic mice model, and embryonic neural progenitor cells to investigate the mutation previously identified in Williams Syndrome. They found abnormal proliferation and differentiation induced by this mutation, as well as altered expression profiles corresponding with aberrant cell clusters. This is regulated through the binding of GTF2IRD1 to transthyretin (TTR) promoter regions and tested on three models mentioned above on neurodevelopmental deficits.

Strengths:

Authors have applied both cell culture, organoid culture and in vivo model to test the previously reported mutation found in Williams Syndrome. They investigated cell behavior including proliferation and differentiation, while using the NGS technique to identify potential signaling pathways that are highly involved and can serve as a candidate to save the phenotype.

---

## [Referee Report · Reviewer #2 (Public Review)]

Summary:

The study by Xingsen Zhao et al on "A human forebrain organoid model reveals the essential function of GTF2IRD1-TTR-ERK axis for the neurodevelopmental deficits of Williams Syndrome" presents a forebrain organoid model for WS and has identified defects in neurogenesis. The authors have performed scRNAseq from these patients' derived forebrain organoids showing upregulation expression in genes related to cell proliferation while genes involved in neuronal differentiation were downregulated. The major findings presented in this study are an increase in the size of SOX2+ ventricular zone in WS forebrain organoids with an altered developmental trajectory and aberrant excitatory neurogenesis. The study also presents evidence that transthyretin (TTR) has a reduced expression in WS organoids, and its expression is regulated by the transcription factor -GTF2IRD1. The authors then go on identity mechanistic details of TTR function on MAPK/ERK pathway which has been known to be involved in brain development. Overall, this is a well-constructed study revealing the function of one of the key genes that is deleted in WS and provides novel insights into mechanisms underlying the abnormal neurogenesis in WS brain.

Strengths:

WS patients have neurocognitive disorders which most likely stem from defects in early neurodevelopment. This study has investigated a WS forebrain organoid model with scRNAseq and identified differences in cell proliferation and differentiation. This study has presented some new evidence regarding the function and regulation of TTR and its regulator GTF2IRD1 during brain development.

Weaknesses:

Though the evidence presented for the mechanism of action of TTR on the MAPK pathway is unclear and lacks depth. It would require identifying downstream targets of TTR and how it interacts with the MAPK pathway.